# Inflammatory and Metabolic Signaling Interfaces of the Hypertrophic and Senescent Chondrocyte Phenotypes Associated with Osteoarthritis

**DOI:** 10.3390/ijms242216468

**Published:** 2023-11-17

**Authors:** Emőke Horváth, Árpád Sólyom, János Székely, Előd Ernő Nagy, Horațiu Popoviciu

**Affiliations:** 1Department of Pathology, Faculty of Medicine, George Emil Palade University of Medicine, Pharmacy, Science and Technology of Targu Mures, 38 Gheorghe Marinescu Street, 540142 Targu Mures, Romania; emoke.horvath@umfst.ro; 2Pathology Service, County Emergency Clinical Hospital of Targu Mures, 50 Gheorghe Marinescu Street, 540136 Targu Mures, Romania; 3Department of Orthopedics-Traumatology, George Emil Palade University of Medicine, Pharmacy, Science and Technology of Targu Mures, 38 Gh. Marinescu Street, 540142 Targu Mures, Romania; arpad.solyom@umfst.ro; 4Clinic of Orthopaedics and Traumatology, County Emergency Clinical Hospital of Targu Mures, 50 Gheorghe Marinescu Street, 540136 Targu Mures, Romania; egyjanek@yahoo.com; 5Department of Biochemistry and Environmental Chemistry, George Emil Palade University of Medicine, Pharmacy, Sciences and Technology of Targu Mures, 38 Gheorghe Marinescu Street, 540142 Targu Mures, Romania; 6Laboratory of Medical Analysis, Clinical County Hospital Mures, 6 Bernády György Square, 540394 Targu Mures, Romania; 7Department of Rheumatology, Physical and Medical Rehabilitation, George Emil Palade University of Medicine, Pharmacy, Science and Technology of Targu Mures, 38 Gheorghe Marinescu Street, 540139 Targu Mures, Romania; horatiu.popoviciu@umfst.ro

**Keywords:** chondrocyte, inflammation, hypertrophy, senescence, metabolic, oxidative stress, NF-kB, YAP/TAZ

## Abstract

Osteoarthritis (OA) is a complex disease of whole joints with progressive cartilage matrix degradation and chondrocyte transformation. The inflammatory features of OA are reflected in increased synovial levels of IL-1β, IL-6 and VEGF, higher levels of TLR-4 binding plasma proteins and increased expression of IL-15, IL-18, IL-10 and Cox2, in cartilage. Chondrocytes in OA undergo hypertrophic and senescent transition; in these states, the expression of Sox-9, Acan and Col2a1 is suppressed, whereas the expression of RunX2, HIF-2α and MMP-13 is significantly increased. NF-kB, which triggers many pro-inflammatory cytokines, works with BMP, Wnt and HIF-2α to link hypertrophy and inflammation. Altered carbohydrate metabolism and the upregulation of GLUT-1 contribute to the formation of end-glycation products that trigger inflammation via the RAGE pathway. In addition, a glycolytic shift, increased rates of oxidative phosphorylation and mitochondrial dysfunction generate reactive oxygen species with deleterious effects. An important surveyor mechanism, the YAP/TAZ signaling system, controls chondrocyte differentiation, inhibits ageing by protecting the nuclear envelope and suppressing NF-kB, MMP-13 and aggrecanases. The inflammatory microenvironment and synthesis of key matrix components are also controlled by SIRT1 and mTORc. Senescent chondrocytes represent the functional end stage of hypertrophic differentiation and characteristically upregulate p16 and p21, but also a variety of inflammatory cytokines, chemokines and metalloproteinases, developing the senescence-associated secretory phenotype. Senolysis with dendrobin, miR29b-5p and other agents has been shown to be efficient under experimental conditions, and appears to be a promising tool for the treatment of OA, as it restores COL2A1 and aggrecan synthesis, suppressing NF-kB and destructive metalloproteinases.

## 1. Introduction

Osteoarthritis (OA) is a complex disease that can affect any joint and involves a series of degenerative, inflammatory and remodeling processes. The early events in osteoarthritis are likely to involve the synovial membrane, which is well-vascularized, has mechanoreceptors and is therefore sensitive to systemic effects. In parallel or in rapid succession, abnormal mechanical loading and other factors may also result in chondrocyte phenotype switching and matrix remodeling; during the course of the disease, all parts of the joint, the articular cartilage, subchondral bone, synovial tissue and meniscus, interact in the development of pathological changes, including cartilage degradation, osteophyte formation, subchondral sclerosis and synovial hyperplasia [1]. The central change is cartilage degradation, which involves both cellular and extracellular phenomena. Inefficient repair contributes to the progression of cartilage deterioration. Chondrogenic progenitor cells are overrepresented in osteoarthritic cartilage compared to the normal joint. In early OA, mobile cell clusters form near the erosions and fissures, showing both anabolic and catabolic phenotypes (Notch-1, Stro-1, and VCAM-1). In advanced disease, migrating islands are distributed throughout the cartilage and can even pass through the fractured tidemark as shuttles between cartilage and subchondral bone. Their putative reparative role during disease progression is not fully understood, and the functional characterization of their subtypes is required to exploit their therapeutic potential [2]. In the course of OA, chondrocytes, the only cells of cartilage, are exposed to non-physiological stresses and biochemical stimuli from matrix degradation products and pro-inflammatory mediators [3]. These cells often undergo phenotypic changes with a shift towards senescence or apoptosis. Chondrocytes, together with synovial fibroblasts and macrophages and subchondral osteoclasts also undergo catabolic reprogramming, leading to matrix degradation and defective remodeling with altered collagen network and microarchitecture [4]. This functional transition is reflected in the presence of two closely related and histologically, immunologically and genetically detectable altered phenotypes, characterized by hypertrophy and senescence, which share several features and often coexist in a stressed environment due to abnormal biomechanical and biochemical challenges [5,6]. As low-grade joint inflammation, characterized by relatively modest numbers of infiltrating cells and soluble inflammatory mediators, is thought to be an important pathway for perpetuating cartilage degradation [7], the aim of this review is to highlight the key inflammatory checkpoints and their interfaces with the hypertrophic and senescent transitions.

## 2. Evidence for the Presence of Low-Grade Inflammation in Osteoarthritis

The inflammation seen in OA differs from that characteristic of rheumatoid arthritis or other autoimmune arthritides. It is a chronic process, of low intensity compared to the aforementioned conditions, and is driven primarily by the innate immune system [7]. There are fewer macrophages and T-cells in OA than in joint tissue affected by rheumatoid arthritis [7]. All parts of the joint are affected: the synovial membrane and synovial fluid, the superficial and deep layers of cartilage, and the subchondral bone. Synovitis is responsible for some of the characteristic clinical symptoms: stiffness, warmth, pain, edema and joint effusion.

Sohn et al. identified 108 proteins of different classes (plasma proteins, serine protease inhibitors, markers of cartilage turnover and inflammatory cytokines) with altered levels in OA synovial fluid [8]. Despite the anatomical compartmentalization of the joint, inflammatory signaling in OA is not restricted to the synovial membrane, but is also characteristic of cartilage and subchondral bone. Disease-associated molecular patterns, such as calcium crystals, extracellular matrix degradation products, HMGB proteins, HSPs and the activation of the complement pathway, can lead to Nod-like receptor, TLR and RAGE signaling and maintain low-grade inflammation [9,10]. Interestingly, several plasma proteins such as Gc-globulin, α1-microglobulin and α2-macroglobulin can enhance pro-inflammatory cytokine production by macrophage cultures via the TLR-4 receptor [8]. Another study comparing synovial protein patterns in early and advanced OA and controls found increased levels of albumin, fibrinogen, alpha1-microglobulin/bikunin precursor, α2-macroglobulin, haptoglobin, ceruloplasmin and other acute phase proteins, and decreased levels of cystatin A and aggrecan 1 [11]. Among the inflammatory mediators, levels of IL-1β and VEGF have been found to be higher in osteoarthritic than in normal sera, while IL-6 can reach 190-fold higher concentrations in OA synovial fluid than in serum [8,12]. CXCL1 can promote IL-6 synthesis in synovial fibroblasts [13]. IL-1β, IL-6, TNFα and VEGF were all significantly elevated in the sera of 55 OA patients with different disease stages compared to controls [14].

In general, animal models support the presence of pro-inflammatory factors in OA, but in some cases there is still some controversy. Experimental OA reflects both primary and secondar, post-traumatic human disease, and includes spontaneous (naturally occurring and due to genetic background), induced (provoked surgically or chemically) and non-invasive murine, canine and lapine models applying a transarticular impact [15]. Meniscal ligament surgery in animals is sufficient to induce NF-kB activation, which is potentiated by IL-1β administration and reflected in high IL-6, MCP-1, MMP-13 and ADAMTS4 synthesis [16]. However, there are OA models in which the classical pathogenic role of IL-1 remains questionable. van Daelen et al. found that IL-1α and IL-1β are not involved in synovial inflammation and cartilage destruction in collagenase-initiated OA [17]. In meniscectomized *IL-1β/NLRP3* double knockout mice induced with hydroxyapatite crystals, the severity and extent of cartilage lesions were even exacerbated compared to their meniscectomized counterparts [18].

In humans, in contrast to the variations of the aforementioned cytokines, changes in osteoarthritic plasma NF-kB and TNF-α levels may be moderate or even lower than in the healthy population [12]. A polyacrilamide gel–liquid chromatography–tandem mass spectrometry proteomics approach identified the presence of several factors and subunits of NF-kB in synovial fluid, such as NF-kB-repressing factor, NF-kB inhibitor-like protein 1, and NF-kB p100 subunit, which are normally found only in the intracellular space [8]. Stannus et al. performed a follow-up study of randomly enrolled subjects with a mean age of 63 years and measured baseline IL-6 and TNF-α levels compared with knee cartilage volume at baseline and after 3 years. The quartiles of IL-6 and TNF-α were associated with medial tibio-femoral joint space narrowing; the baseline IL-6 levels and change in IL-6 levels predicted the loss of medial and lateral cartilage volume [19]. In the study by Barker et al., TNF-α was lower in mild OA but not different in advanced OA than in controls, whereas IL-10 and the IL-10/TNF-α ratio were significantly lower in those with primary unilateral anterior cruciate ligament surgery and those who underwent total joint replacement [20]. In another human study, plasma leptin levels and synovial IL-18 were associated with Kellgren–Lawrence score [21]. In a small study group, Waszczykowski et al. observed increased serum levels of IL-18 and IL-20 compared to controls, with a significant correlation between IL-18 and MMP-3 levels [22]. IL-17 deficiency improved pain and cartilage degradation in *IL-1Rα/IL-17* double knock-out mice with chemically (MIA)-induced OA [23]. The characteristic inflammatory changes seen in the synovial fluid and the bloodstream are represented in Figure 1.

Circulating cytokines can have many cellular sources, so there may be significant interference and high biological variability in their serum or plasma levels. They can originate from synovial macrophages and fibroblasts, subchondral osteoblasts, osteoclasts and chondrocytes, but also from other organs, with additive, synergistic or antagonistic effects on chondrocyte function. It is not easy to quantify locally produced and delivered samples, and powerful morphometric and imaging techniques are required for such tasks.

In terms of histological assessment, the expression of IL-1β, IL-6, TNF-α and VEGF proteins in the synovial membrane was significantly higher in moderate and advanced OA than in mild OA [14]. IL-6 appears to play a role in perpetuating inflammation, as it induces the deposition of hydroxyapatite crystals in human cartilage explants, providing a feedback loop for its synthesis [24]. Qu et al. found higher IL-6 and MMP-9 gene and protein expression in joint replacement surgery or arthroscopy specimens from OA patients compared to healthy controls [25]. Osteoarthritic synovial samples and LPS-induced synoviocytes showed upregulation of the nucleotide-binding oligomerization domain-like receptor pyrin domain containing 3 (NLRP3) inflammasome, in parallel with the redox homeostasis transcription factor Nrf2 and heme oxygenase-1 [26]. IL-1β and IL-18 were also overexpressed and decreased when Nrf2, but not NLRP3, was knocked down [26]. Warner et al. found that IL-15Rα was expressed in osteoarthritic cartilage samples, and IL-15 induced MMP-1 and MMP-3 synthesis in cultured cartilage explants. Genetic polymorphisms have been associated with the risk of symptomatic versus asymptomatic OA, and with the risk of neuropathic pain-like symptoms after total joint replacement surgery [27]. IL-15 was elevated in synovial fluid and synovial membrane cells obtained by arthroscopy in patients with early knee OA [28]. Counterregulatory mechanisms are also amplified, as high levels of IL-10 mRNA and protein have been detected in high-intensity cartilage lesions [29].

IL-1β and IL-6 trigger the cyclooxygenase pathway, and selective Cox-2 inhibition has raised high hopes as a candidate for disease-modifying drugs. Indeed, there is some experimental evidence that in chemically induced animal OA, meloxicam, a preferential Cox-2 inhibitor, has beneficial effects in inhibiting collagen type II degradation and cartilage destruction [30,31]. One possible trigger of the cyclooxygenase pathway is leptin: a concentration of 10 μg/mL increased the expression of Cox-2 and IL-6 as well as MMP-1, -3, -13, whereas suppressor of cytokine signaling-3 (SOCS-3) was able to stop these effects [32]. Serum leptin levels are elevated in obese OA patients, especially those with a high body-mass index (BMI), compared to non-OA, non-obese, and osteoarthritic individuals with low BMI, and correlate with the radiographic stage of the disease [33]. Moreover, 10 μg/mL leptin triggering alone, or with simultaneously administered IL-1β (10 pg/mL), stimulated the expression of inducible nitric oxide synthase, PGE_2_, IL-6 and IL-8 in femoral condyle and tibial plateau cartilage specimens obtained in the event of joint replacement [34].

The evidence for the presence of inflammatory mediators in blood, synovial fluid and joint histological samples is summarized in Table 1.

## 3. Sox-9, NF-kB and Proinflammatory Cytokines Mediate Hypertrophy and Aberrant Signal Transduction Leading to Cartilage Deterioration

In early OA, in addition to Runx2, chondrocytes upregulate HIF-2α and show characteristically increased expression of Sox-9, Col2a1 and ACAN [35]. Sox-9 is a transcription factor belonging to the HMG-box class of DNA-binding proteins that controls the overall chondrogenesis process and is downregulated during hypertrophic transformation. There are many signaling pathways that modulate Sox-9 at multiple levels, such as inflammation or epigenetic factors. IL-1β induces the MAPK/ERK/JNK kinases, of which ERK inhibits collagen type II and aggrecan synthesis, while JNK represses Sox-9 [36,37]. Sox-9 exerts downstream effects through the AMP-response-binding protein (CREB)/p300 [34], and is also modulated by IKK-β/NF-kB, acting in parallel with HIF-2α and the BMP signaling system [38]. The Transient Receptor Potential Vanilloid 4 (TRPV4), an ion channel found by functional gene screening to play a pivotal role in chondrocyte mechanotransduction, stimulates Ca^2+^ influx, Sox-9, and matrix synthesis [39,40]. TRPV4 is a regulatory checkpoint that also switches off also the IL-1 signaling through the downregulation of IL-1R [41]. Sox-9-driven transactivation is epigenetically controlled by several enzymes: histone acetyltransferases, methyltransferases ESET and EZH2, and the demethylase Phf2 [42,43,44]. HDAC4, a NAD+-independent histone deacetylase, triggers Sox-9 and the hypertrophic transition, as observed by Dong Z. et al. in a longitudinal study of *ACAN-CreERT2-HDA4 fl/fl* transgenic mice with early, age-related OA with synovitis and osteophyte formation; HDAC4 silencing resulted in a very significant increase in RunX2, Col10a1, IHH signaling and MMP-13 expression, whereas Sox-9, ACAN and Col2a1 were overtly suppressed [45]. SIRT1 and SIRT6 exert important downstream regulation: the former increases COL2A1 and ACAN expression in the healthy state, while the latter suppresses several inflammatory genes. Both SIRT1 and SIRT6 are downregulated in OA [46]. Some controversies remain, as a previous study reported the decrease in Sox-9 expression in parallel with COL2A1 and ACAN in advanced OA [47].

Osteoarthritic chondrocytes are chronically exposed to oxidative stress and inflammatory stimuli that induce their transformation into a fibroblast-like and hypertrophic phenotype [35]. These reprogrammed chondrocytes, synovial fibroblasts and macrophages, but also subchondral osteoclasts, are the source of IL-1β, IL-6, TNF-α and leptin, and induce the synthesis and secretion of MMP-2, -3, -9 and -13 and ADAMTS5, leading to intense extracellular matrix destruction. The overt linkage between inflammation and degradative pathways also has experimental evidence; the lipopolysaccharide conditioning of chondrocyte culture media induced intensive NF-kB, IL-1β, and MMP-13 synthesis, associated with reduced viability. All of these expressions can be suppressed by antioxidant mixtures such as ascorbic acid, β-caryophyllene and D-glucosamine [48]. β-caryophyllene couples with the cannabinoid receptor CB2, and enhances the expressions of PPAR-γ and PGC-1α, which may mediate its flagrant anti-inflammatory effects [49]. Chondrocytes adjust their signaling and respond by upregulating four signaling pathways, Wnt, NOTCH, HIF-2α and NF-kB, all of which provide a link between inflammation and hypertrophy [50]. At the effector level, these factors converge to address TNF-α and MMP-13 production via the p38MAPK and JNK/Runx2 signaling axes [51,52]. Among the above pathways, the canonical NF-kB pathway seems to play a central role, collaterally triggering pro-hypertrophic BMP2, the growth arrest and the DNA-damage-inducible 45 (GADD45) factor [53]. Canonical and non-canonical NF-kB signaling is an essential conductor in the hypertrophic transformation, and also orchestrates the progression of osteoarthritis [54], acting under the control of IKK-α and IKK-β kinases [55]. In addition to NF-kB, BMPs also appear to be important, as in a cultured microcartilage model, the selective inhibition of BMP receptor type I abolished some hypertrophic features, such as MMP-13 and DIPEN, a C-terminal aggrecan neoepitope generated by MMP-13 [56].

## 4. Hypertrophic Differentiation and Chondrocyte Phenotype

Chondrocyte hypertrophy is a key feature of endochondral ossification, cartilage and bone development [57], and in the early stages of OA, dedifferentiated chondrocytes resemble their precursors in terms of proliferative and synthetic capacity. Hypertrophy represents terminal chondrocyte differentiation. Hypertrophy-like changes characterize both human and experimental OA, but do not involve all cells of the cartilage in a synchronized manner [58]. It has been proposed that hypertrophic differentiation of healthy articular chondrocytes occurs, at least in part, as a result of water entry through aquaporin channels [59]. The cells show increased organelle synthesis [57], and soluble compounds, and also the organic, stable microenvironment, interact with the differentiation process. In cell culture monolayers, regardless of their origin, chondrocytes lose their usual morphology and reorganize their cytoskeleton into an intensive production of thick actin fibers. Furthermore, the nuclear localization of myocardin-related transcription factor (MRTF) and increased expression of type I collagen (*Col1*) and tenascin C (*Tnc*) genes were observed, which was associated with the downregulation of Sox-9 gene expression [60].

Once differentiated during endochondral ossification, hypertrophic chondrocytes synthesize several metalloproteinases such as MMP-13, the gelatinases MMP-2 and MMP-9, and the stromelysins MMP-3 and MMP-10 [61]. In OA, hypertrophic chondrocytes indirectly mediate matrix remodeling by controlling chondroclasts, the inhabitants of the hyaline cartilage erosion zone [57]. These cells share morphological and transcriptomic features with osteoclasts, but are distinct in some features, with higher expression of the *P2rx5*, *Nxn*, *Gspt1*, *Serp1*, *Scrn1* and *Rac1* genes, but lower levels of ACP5 (tartrate-resistant acid phosphatase) [62]. Hypertrophic cartilage differs from normal cartilage in terms of the organization of both the cellular and extracellular matrix architecture. The superficial layer is disorganized, with focal matrix condensations, erosion and frequent matrix fibril ruptures. A characteristic glucosaminoglycan depletion can be seen in the deeper layers, which can be detected by safranin O staining [59]. In healthy cartilage, chondrocytes are homogeneously distributed, flattened in the superficial layer but round in the deep region. In contrast, osteoarthritic cartilage shows aggregated islands of chondrocytes of variable shape with the loss of characteristic organization and polarity. Experience gained in bioengineering experiments has shown that it is difficult to obtain a stable articular cartilage construct because of the shift to the so-called transient phenotype and hypertrophy instead of a stable permanent structure [59]. In the course of mesenchymal stem cell differentiation, several groups of signaling molecules dominate; BMP2,4,6, Indian hedgehog (IHH), parathormone-related peptide (PTHrP) signaling and the presence of angiogenic VEGF and pro-inflammatory IL-1β reflect commitment to the transient phenotype [59]. This transient cartilage contains high levels of collagen-I (COL-I), collagen-X (COL-X) and RUNX2, which confer a bone-like structure, and its final differentiation does indeed result in bone. In comparison, normal healthy articular chondrocytes characteristically express Sox-9 and COL-II, and GDF-5, autotaxin (ATX) and Chordin in their environment, induced mainly by the Wnt agonists Wnt4 and Wnt9a [63]. The phenotype, signaling, and secretory features of hypertrophic chondrocytes are represented in Figure 1.

The question that arises is whether chondrocyte hypertrophy is an obligatory feature of osteoarthritis. It appears that experimental hypertrophy is not always associated with OA, but a variable number of hypertrophic chondrocytes can be found in osteoarthritic cartilage.

Catheline et al. used the chondrocyte-specific overexpression of the transcription factor RunX2 with a *Rosa26Runx2* allele, combined with inducible transgenes for COL2A and ACAN, and found that despite the early onset of chondrocyte hypertrophy, RunX2 overexpression alone was not sufficient to trigger OA. However, in a model of knee joint destabilization and meniscal injury, they described progressive cartilage degeneration characterized by impaired OARSI scores, increased apoptosis and high MMP-13 expression [64].

At least four activated signaling systems converge at the level of RunX2, distinguishing the signalomic signatures of normal and hypertrophic chondrocytes: a. the Wnt/LRP5/6/Fzd/β-catenin, b. the IHH/Ptch1, c. the TGF-β/TGFR/SMAD2,3,4, and d. the BMP2,4/SMAD1,5,8 [50]. Wnt ligands may play a dual role in the regulation of RunX2; Wnt 3, 8, 9 bind to the Frizzled receptor and enhance the nuclear translocation of β-catenin, and consequently the expression of RunX2 [59,65], triggering hypertrophy, whereas the non-canonical agonist Wnt 5a initially activates but later suppresses it by inhibiting RunX2 [66]. Ferrao Blanco et al. reported interesting results on the hypertrophic phenotype of human articular chondrocytes induced by inflammatory cytokines or medium conditioned by defined macrophage subsets. They found that genes associated with hypertrophy (Col10a1 and RunX2) were downregulated by the mixture of cytokines IL-1β, TNF-α and IFN-γ in cartilage explants and chondrocytes packed in alginate, and only MMP-13 expression could be upregulated by these stimuli. NF-kB inhibition with SC-514 did not significantly alter the expression of Col10a1, RunX2 and MMP-13, nor did the addition of medium-conditioned TNF-α/IFN-γ producing macrophages. On the contrary, IL-4-synthesising reparative macrophages were able to upregulate COL10A1, RUNX2 and IHH in cartilage explants, which the authors considered to be more important than the intrinsic inflammatory signals [67]. The role of macrophages in OA is also supported by the finding that macrophage depletion in mice protects against collagenase-induced OA [68].

Due to abnormal mechanical loading, excessive oxidative stress and other factors, the normal chondrocyte in OA undergoes hypertrophic transformation. In addition to increased size, this phenotype is manifested by the overexpression of Sox-9, RunX2 and COL10A1 and the suppression of COL2A1 and ACAN. The Wnt, IHH, TGF-β and BMP-2 signal pathways are amplified, and pro-inflammatory cytokines and several metalloproteinases such as MMP-2, MMP-3, MMP-9 and MMP-13 appear in the secretory pattern. Osteoarthritic cartilage shows low-grade inflammation and the degradation of the extracellular matrix (collagen type II, aggrecan and glycosaminoglycan components). The accumulation of microcrystals favors the transformation, and they have a mutual triggering effect with IL-6. The final differentiation of hypertrophic chondrocytes results senescent cells, characterized by the overexpression of P16 and p21, increased RunX2 and the newly discovered HMGB2, NDRG2 and TSPYL2. The senescence-associated secretion pattern includes IL-1β, IL-6, IL-8, MCP-1, VEGF, MMP-1, MMP-3 and MMP-13. As elevated levels can be detected in the bloodstream, some of these can be considered potential biomarkers for OA. Some cytokines, such as IL-6 and IL-15, also show characteristic elevations in synovial fluid.

## 5. Altered Carbohydrate Metabolism Is Linked to Catabolic Reprogramming with Common Features of Pro-Inflammatory Phenotype in Osteoarthritic Chondrocytes

The population of normal, healthy chondrocytes has a high individual metabolic rate but a low metabolic rate relative to the volume of cartilage. In hyaline cartilage, chondrocytes represent only 1–5% of the total volume [5]. They synthesize and remodel the molecules of their microenvironment: aggrecan, glycoproteins, hyaluronic acid, collagen type II alpha 1 (COL2A1) in the superficial cartilage layer, collagen types IX and X in the deep zone, collagen type VI in the pericellular region. The master regulator of the synthesis of the dominant matrix components, collagen and ACAN, is SOX-9 [5].

Cartilage is an avascular and alymphatic tissue, and chondrocytes have limited access to oxygen and essential nutrients such as glucose or amino acids. During maturation, chondrocytes undergo bioenergetic reprogramming involving glycolysis, oxidative phosphorylation, the pentose phosphate cycle [69] and glycogen metabolism. They possess GLUT-1 and GLUT-3 transporters to ensure glucose uptake, of which GLUT-3 is constitutively expressed, but GLUT-1 expression is highly adaptable to the limited glucose supply [70]. The sustained high expression of GLUT-1 and increased glucose uptake may, in turn, facilitate cartilage destruction through the accumulation of end-glycation products and proteoglycan consumption [71]. End-glycation products are stable macromolecular complexes formed by non-enzymatic glycosylation reactions, often at lysin or arginin residues of matrix proteins. They lead to the formation of new covalent bonds, oxidize thiol groups, inappropriately cross-link collagen fibers and facilitate the formation of protein aggregates. RAGE appears to induce chondrocyte hypertrophy by triggering pro-inflammatory cytokines. In vitro RAGE signaling mimics hypertrophy; S100A11, a recombinant ligand of the receptor for advanced glycation end products, induced an increase in the size of cultured human chondrocytes and the overexpression of p38 MAPK and collagen type X, hallmarks of hypertrophy. On the other hand, hypertrophy induced by TNF-α and CXCL8 was reversible after treatment with soluble RAGE or RAGE-specific antibody blockade [72].

Energy-harvesting pathways are closely linked to extracellular matrix turnover in cartilage. In osteoarthritic chondrocytes, anaerobic glycolysis dominates, with low rates of energy production [73]. Chemical inhibitors of glycolysis, such as iodoacetate (MIA), are sufficient to induce experimental OA [30,74], but aerobic glycolysis is also enhanced as part of the Warburg effect [75]. Glycolysis provides a significant proportion of energy requirements in an oxygen-rich environment, and several glycolytic enzymes have been found to show gene expression changes in OA; hexokinase 2 (HK2) and lactate dehydrogenase A (LDH A) are upregulated in synovial tissue and macrophages, while 6-phosphofructose 2-kinase/fructose 2,6-bisphosphatase 3 (PFKFB3) is downregulated and pyruvate kinase M2 (PKM2) is upregulated in chondrocytes [76]. All these enzymes control additional metabolic processes, and it appears that the main metabolic regulatory principle is different from energy release. The expression of the key glycolitic regulatory enzyme 6-phosphofructose-2-kinase/fructose-2,6-bisphosphatase 3 (PFKFB3) is suppressed in osteoarthritic chondrocytes [77]; in healthy chondrocytes, this enzyme ensures normal COL2A and aggrecan synthesis via phosphoinositol 3-kinase (PI3K)/protein kinase B (AKT)/C/EBP homologous protein (CHOP), but inhibits caspase-3 activity and consequent apoptosis [77]. Another allosteric enzyme of glycolysis, pyruvate kinase (PK), which catalyzes the final, irreversible conversion (phospho-enol-pyruvate to pyruvate) of this reaction series, also has a relevant role in reducing SOX-9 and COL2A production, apoptosis and favoring proliferation [78]. PKM2 enhances the Warburg effect [73]. Finally, lactate accumulation may have a dual role. First, lactate increases the lactylation of histone lysine residues, suppressing the activation of M1-type macrophages and stimulating polarization to the reparative M2-type [79]. On the other hand, the conversion of pyruvate to lactate by lactate dehydrogenase under anaerobic conditions triggers nuclear factor kappa B (NF-κB) signaling, which can induce the expression of pro-inflammatory cytokine genes and further reprogram energy metabolism in chondrocytes [16]. Furthermore, LDH A and B have been shown to enhance ROS production in HeLa and cholangiocarcinoma cells, increasing hydrogen peroxide and superoxide generation in mitochondria [80]. However, in lymphoma B-cell lines, the inhibition of LDH A with siRNA resulted in increased oxygen consumption and ROS generation by diverting lactate towards pyruvate recovery and the oxidative phosphorylation pathway [81]. Based on these observations, targeting glycolysis appears to be a promising approach to the treatment of OA [76]; however, delivering a tissue-specific drug with limited, local effects will be challenging.

In addition to glucose, glycogen metabolism is abnormally reduced in osteoarthritic cartilage. Glycogen is required for joint tissue homeostasis, and glycogen synthase, the key enzyme in glycogen synthesis, is inhibited when it is phosphorylated on several serine residues. Phosphorylated glycogen synthase kinase was found in the deep cartilage layer in osteoarthritic specimens, and thus lower glycogen levels were probable; in contrast, the two isoforms of the enzyme are unphosphorylated and active in normal tissue [82].

If chondrocytes derive their energy mainly from glucose, with regard to alternative nutrients, little is known about the direct effects of lipid and amino acid metabolism on hypertrophy and/or senescence. However, these substances interfere with inflammation, cartilage degeneration and remodeling at several checkpoints. First, there is impaired β oxidation in the mitochondria and a reduced carnitine shuttle, which is essential for the transfer of various activated acyl groups. Saturated fatty acids such as stearate and palmitate are pro-inflammatory, the former increasing cytokine production via HIF-1α and the latter inducing Cox-2 and IL-6 [73]. N-6 polyunsaturated fatty acids increase ROS production by stimulating NADPH + H^+^ oxidase 4, promoting apoptosis, and arachidonic acid contributes to the generation of pro-inflammatory prostaglandins and leukotrienes [73]. N-3 polyunsaturated fatty acids are strongly anti-inflammatory, as they form pro-resolving mediators such as resolvins, protectins and maresins, and suppress ADAMTS-4, ADAMTS-5, MMP-3 and MMP-13 [83]. In addition, osteoarthritic chondrocytes increase their cholesterol uptake and produce increased amounts of oxysterols [84]. Phospholipid degradation products, such as lisophospholipids produced by phospholipase A2 (PLA2), recruit phagocytes into the joint. Many quantitative changes in amino acids have been proposed as biomarkers, such as an increase in arginine/asymmetric dimethylarginine or a decrease in the ratio of branched-chain amino acids/histidine. In synovial fluid, glutamine, threonine and nitrotyrosine reach high concentrations; increased proteolysis and nitrosative stress may contribute to these metabolic changes [73]. Arginase 2, an important enzyme of infiltrating macrophages, upregulates MMP-3 and MMP-13 via NF-kB [85].

## 6. Central Metabolic Pathways’ Reprogramming and Mitochondrial Dysfunction Are Hallmarks of Hypertrophy and Senescence

The metabolic reprogramming of osteoarthritic chondrocytes also involves the central metabolic pathways: the tricarboxylic acid (TCA) cycle and oxidative phosphorylation (OXPHOS) [73]. The TCA consists of nine reactions, three of which are irreversible and produce CO_2_ and reduced equivalents (NADH + H^+^ and FADH_2_). OXPHOS directs NADH + H^+^ and FADH_2_ to re-oxidation in the inner mitochondrial membrane, relying on five enzyme complexes, resulting in H_2_O. The substrate input into the TCA is reduced in OA as less pyruvate is converted to acetyl-CoA [86]. However, the three key regulatory enzymes, citrate synthase, isocitrate dehidrogenase and α-cetoglutarate dehidrogenase, are upregulated [73].

Inflammatory mediators interfere with these reactions; for example, IL-1β expression, which is increased in both experimental and human OA, suppresses isocitrate dehydrogenases 1 and 2 in chondrocytes [87]. Other authors found impaired mitochondrial function with reduced IDH activity in an endemic degenerative form of OA, Kashin–Beck disease [88].

Wang Y et al. observed that reduced mitochondrial biogenesis and oxidative phosphorylation (OXPHOS) are associated with the reduction in AMPKα and PGC1-α activity, and can be reversed by selective pharmacological activators that increase AMPK phosphorylation [89]. Knockdown of mitochondrial transcription factor A was associated with a marked inhibition of respiratory complex III, especially in the presence of high IL-1β concentrations (10 ng/mL). IL-1β treatment causes metabolic reprogramming and a Warburg-like effect first described in malignant cells. In mice, high-dose IL-1β treatment (10 ng/mL) increases the rate of glycolysis but decreases basal and maximal respiration, the intensity of the citrate cycle (TCA), and the expression of Glut-1, G-6PD2 and LDHA effects that are abolished by the deletion of IKK [54]. From a biochemical point of view, in addition to the inhibition of TCA and OXPHOS, IL-1β suppresses fatty acid metabolism and the degradation of aliphatic branched amino acids, but significantly stimulates glutathione synthesis, an important antioxidant pathway [54]. On the other hand, LDHA induced by IL-1β triggers many catabolic genes and the IKB-ζ protein, and its binding to NADH + H^+^ generates reactive oxygen species. IKB-ζ activates the downstream regulator RANKL and the NLRP3 inflammasome, the promoter of IL-1β activation [16].

An important factor in maintaining the balance between glycolysis and OXPHOS is IGF2. Hypertrophic chondrocytes from IGF2^−/−^ mice enhance their OXPHOS and pentose phosphate pathways [90]. Protein and lipid components, as well as nucleotide precursors, are required to ensure the anabolic boost of the hypertrophic state [90,91]. The first, oxidative phase of the pentose phosphate pathway produces NADPH + H^+^ and pentoses, which can be incorporated into nucleic acids; when the second, non-oxidative phase also occurs, only NADPH + H^+^ remains as the sole reaction product. NADPH + H^+^ is an electron and proton donor, a strong reducing agent that neutralizes free radicals, which is supported by the stimulation of glucose-6-phosphate dehydrogenase (G6PDH) activity in chondrocyte hypertrophy [69]. Thus, these mechanisms inhibit the excessive oxidative stress generated by the deficient activity of complexes II and III, the reduction in the inner membrane potential and the consequent mitochondrial damage during the hypertrophic transition.

Mitochondrial dysfunction has been described to be directly related to cellular senescence. In adipocytes and keratinocytes, mitochondrial dysfunction-associated senescence (MiDAS) is induced, characterized by low NAD+/NADH ratios and an altered SASP pattern controlled by the NAD+/AMPK/p53 pathway, with CCL-27, TNF-α and IL-10 secretion without the involvement of NF-kB and IL-1β [92,93].

Chondrocytes grown on U937 monocyte-derived macrophage-conditioned medium synthesize high-levels of IL-1β, IL-6, TNFα and the mitochondrial ROS. Hyaluronic acid, lactose-modified chitosan and their mixtures significantly reduced these effects [94]. Twenty-four-hour treatment with 10 ng/mL recombinant IL-1β and heterogeneic bone crystal microparticles (obtained from long cortical porcine bones via mechanical shredding, sterilization and a 70% alcohol wash) also induced mitochondrial dysfunction, with increased ROS levels due to increased electron transport chain activity [16]. This oxidative stress can progress to apoptosis; when induced chondrocytes were treated with the electron transport chain inhibitors rotenone and antimycin A, IκB-ζ protein, IL-6 and MMP-13 levels decreased [16]. When nitrosative stress was mimicked with an NO donor, S-nitrosoglutathione, a significant increase in p21 but not p16 expression was observed [16], promoting a senescent transition. Guidotti et al. highlighted an important link between canonical Wnt signaling, oxidative damage and cellular senescence. Using chondrocytes from cartilage explants from obese patients, they blocked GSK3, a key component of the β-catenin phosphorylation complex, with lithium chloride. This had remarkable consequences: it increased oxidative stress and DNA damage, induced GADD45β and p21, and increased the number of SA-β-galactosidase-positive cells [95].

## 7. Imbalance of Hypoxia-Inducible Factors Regulate Inflammatory Signals and Cartilage Destruction

As an avascular and alymphatic tissue, cartilage is adapted to chronic hypoxic conditions and relies on diffusion to supply organic nutrients to the chondrocytes. The hypoxia regulatory factor HIF-1α is constitutively produced under conditions of normal oxygen supply. The further induction of HIF-1α suppresses the canonical Wnt pathway via β-catenin and RunX2, but promotes the effects of Sox-9 [96]. There is an O2 gradient in cartilage with a maximum of 6% in the superficial layers and a minimum of <1% in the deep zone [73]. Human articular chondrocytes, when cultured under hypoxic conditions (2% O2) in monolayers or 3D pellets, retained their phenotype by upregulating Sox-9, Frizzled-related protein (FRZB) and COL2A1, and showed fewer markers of dedifferentiation into mesenchymal stromal cells (CD73, CD90 and CD105) [97]. Chondrocytes are sensitive to higher levels of mechanical forces coexisting with hypoxia (1% O2). These conditions can induce chondrocyte apoptosis by activating endoplasmic reticulum stress [98]. When mechanical overload was applied in the absence of hypoxia, both HIF-1α and HIF-2α were induced and subsequently suppressed by the ER stress inhibitor salubrinol, but HIF-1α showed greater changes [98].

HIF-1α increases glycolysis by stimulating the GLUT-1 transporter and phosphoglycerate kinase 1 (PGK1) and slows down mitochondrial respiration [82]. The results regarding the alterations of HIF-1alpha in OA are equivocal. In experimental OA, HIF-1 is downregulated, and its selective knockdown in chondrocytes favors cartilage destruction [99]. HIF-1 has several important interactions with inflammatory factors. First, it inhibits NF-kB signaling and maintains cartilage structure [100]. Second, HIF-1α also cooperates to support the activity of microsomal prostaglandin E synthase 1 (mPGES-1), and another study reported that HIF-1α is upregulated in OA chondrocytes together with mPGES-1 (possibly through a counterregulatory mechanism) [101]. Lee et al. used N-phenyl-N’-(4-benzyloxyphenoxycarbonyl)-4-chlorophenylsulfonyl hydrazide (PBCH) to inhibit microsomal prostaglandin E synthase-1 (mPGES-1) in an experimentally induced rat model of inflammatory arthritis, and observed that it reduced paw edema and the RANKL/OPG ratio [102].

The initiation of multiple pro-inflammatory cytokines by NFkB shifts the balance to HIF-2α, which triggers VEGF and MMP-13 gene transcription, along with aberrant collagen expression. In addition, IL-1β amplifies these effects through a positive regulatory loop by phosphorylating c-Jun N-terminal kinase (JNK) and further stimulating HIF-2 [103]. Other metalloproteinases, including MMP-1, MMP-3, MMP-9 and effector enzymes such as ADAMTS-4 and NOS2, are also induced and contribute to extracellular matrix destruction [104]. In conclusion, there is a higher prevalence of HIF-2α than HIF-1α in OA cartilage, which is responsible for triggering matrix degradation.

## 8. Senescent Chondrocytes Possess Remarkable Pro-Inflammatory and Catabolic Signatures

As a process, cellular senescence appears with the loss of replicative capacity, observed in cell culture experiments such as the Hayflick effect, or when cultured cells obtained from elderly subjects have a shorter life span than those obtained from young subjects due to telomere shortening and associated chromatin changes [105]. Two different types of senescence have been described, “replicative” senescence and a second form, called “stress-induced” or “extrinsic” senescence, which is associated with oxidative-stress-induced DNA damage and chronic low-grade inflammation. Aging itself, increased biomechanical stress and the depletion of antioxidant systems all contribute to the generation of increased ROS, such as peroxynitrite, superoxide and nitric oxide free radicals [105,106]. An important source of ROS is NADPH oxidase, whose activity is increased in the context of catabolic programming. IL-1-, TNF-α-, FGF- and TGF-β-triggered chondrocytes can produce high levels of ROS [106], and these reduce proteoglycan synthesis, as shown in chondrocyte cultures stressed with H_2_O_2_ [107].

Chondrocyte senescence is thought to be a functional end stage of hypertrophic differentiation and is associated with obvious morphological changes: increased size, flattening and vacuolization. Functionally, the dominant features of senescent chondrocytes are growth arrest, resistance to cell death and the existence of a senescence-associated secretory phenotype (SASP) [108], which is manifested by the expression of many inflammatory genes, such as IL-1β, IL-6, IL-8, MCP-1, TGF-β, VEGF, GRO-α, IGFBP7 and several metalloproteinases along with their inhibitors, MMP-1, -3, -10 and -13 and TIMP1 [108]. Of these factors, IGFBP7, IL-6, IL-8 and also PAI-1 tend to induce senescence either by autocrine or paracrine mechanisms [108].

In the senescent state, a few regulators of the cell cycle are characteristically overexpressed: cyclin-dependent kinase inhibitor 2A (CDKN2A or p16), CIP1/WAF1 (p21) and the tumor suppressor protein p53. Chondrocytes with high p16 expression also express pro-inflammatory cytokines IL-1, IL-6 and matrix metalloproteinases, but are low synthesizers of COL2A [5,6]. Senescence can also occur with DNA damage and consequent growth arrest. Common features of hypertrophic and senescent chondrocytes have been identified, often coexisting in a high-stress microenvironment. High RunX2 expression has been described in both cell types, and RunX2 deletion in a surgically destabilized knee OA animal model results in increased levels of MMP-9 and -13 and ADAMTS-4, -5, -7, and -12 [5]. A short synthesis of the senescent phenotype is shown in Figure 1.

A proteome catalogue of 62,449 chondrocytes from healthy and osteoarthritic subjects, using published single-cell RNA-sequencing datasets, identified clusters of key osteoarthritis proteins. The authors of this study propose that TSPYL2, a DNA repair factor, HMGB2, a modulator of DNA bending and flexibility, and NDRG2, a protein that inhibits the DNA binding ability of NF-kB p65, may be the key regulators of cellular senescence in articular chondrocytes [109].

NF-kB is a key regulator of the senescence-associated secretory phenotype (SASP) 1. In addition to blocking mTOR [110], rapamycin inhibits NF-kB signaling and membrane-bound IL-1β, thereby mediating an inhibitory effect on IL-6 and IL-8 secretion [111]. As oxidative stress induces the NF-kB pathway and ultimately facilitates the release of MMPs, this inhibition disrupts the harmful signaling loop between reactive oxygen species and macromolecular matrix components.

In OA, at least two signals, inflammatory levels of IL-1β and bone mineral particles, promote NF-kB activation. Primary mouse chondrocytes, cultured in DMEM, treated with 10 ng/mL recombinant IL-1β produce the receptor activator of NF-KB (RANKL), which determines osteoclastogenesis in subchondral bone. Bone mineral particles are released into the synovial fluid from cartilage erosion, osteophytes and subchondral osteoclasts. Both IL-1β and bone particles induce signatures of senescence, such as p16 gene expression, and the SASP pattern, characterized by IL-6, Lcn2 and MMP-13 expression [16]. Senolytic treatment with dasatinib and quercetin downregulated these genes, even when IL-1β and bone particles were maintained, and partially reversed the decrease in aggrecan synthesis [16]. Direct evidence that NFKB-driven inflammation and senescence are linked is provided by the overexpression of p16 in chondrocytes, where the constitutively active IKB kinase is also upregulated [16]. Since the specific SASP pattern in IκB-ζ-deficient chondrocytes cannot be induced by the aforementioned factors, it appears to be a central regulator of the SASP response [16].

SIRT-6, a protein deacetylase and ADP-monobiosyltransferase enzyme, deacetylates STAT5, stops its translocation to the nucleus and thus silences IL-15/JAK stimulation. SIRT6 is downregulated in human osteoarthritic cartilage and chondrocytes, its short hairpin RNA knockdown worsens OARSI cartilage damage scores, and its overexpression attenuates p16lnk4a expression, senescence and surgically induced experimental OA. Site-directed mutagenesis of STAT5 (Lys163Arg) abolishes its protective effect [112]. A selective allosteric activator (MDL-800, encapsulated in polyamidoamine) of SIRT-6 has been proposed to silence JAK3/STAT5 dependent transcriptional activity, IL-15 inflammatory signaling and senescence in OA [112].

## 9. Mechanosensing, the YAP/TAZ Signaling, Oxidative Stress and Chondrocyte Senescence

The Yes-associated protein, YAP, and its transcriptional co-regulator, TAZ, are actors in the morphogenetic Hippo signaling pathway, which was originally described as a major regulator of organ size [113]. In the ‘ON’ function, Hippo signaling involves MST1/2, SAV, LATS1/2 and MOB1A/B kinases, which ultimately lead to the phosphorylation of YAP/TAZ, with consequent sequestration in the cytoplasm and loss of its transcriptional activity. During development, YAP activation in *Mob1a/b^−/−^* mice leads to impaired chondrocyte differentiation and chondrodysplasia [114]. YAP/TAZ promotes chondrocyte proliferation but inhibits chondrogenic differentiation, binds to key transcription factors and interacts to reduce canonical Wnt/β-catenin signaling [115,116]. Other work has shown that high YAP/TAZ induces dedifferentiation via RhoA [117], and that it has lower nuclear expression in hypertrophic chondrocytes [114].

There are reciprocal interactions between YAP/TAZ and inflammatory mediators and effectors of cartilage degradation. Overexpression of *Yap* in *Mst 1/2* mutant mice was associated with lower levels of MMP-13 and reduced inflammation in ACLT (anterior cruciate ligament transection) and DMM (destabilization of the medial meniscus) animal models. TNF-α treatment of these mutant chondrocytes significantly reduced YAP expression, along with Col2A1 and aggrecan, and increased MMP and ADAMTS 4/5 expression [118]. Furthermore, overexpression of YAP suppressed NF-kB and decreased phosphorylation of the p65 subunit, even under TNF-α treatment, due to inhibition of IKKα/β and TAK1 [118]. Direct overexpression of YAP in *Col2a1-Yap1tg/+* showed good cartilage integrity in untreated animals and a high level of cartilage resistance under osteoarthritic conditions [118]. These results showed that YAP is a dominant factor in cartilage homeostasis [119], and several experiments performed on different tissues may shed light on the molecular background of these effects.

Expression of YAP-S127A, a phosphorylation mutant of YAP, in fibroblasts and vascular smooth muscle cells stops ageing. In contrast, cells with YAP/TAZ deletion show overt signs of ageing. A possible underlying mechanism may be the protection of the integrity of the nuclear envelope, mediated by the normal synthesis of lamin A, lamin B1 and actin-related protein 2 (Actr2), components of the nuclear lamina, which tightly envelops the materials of the nucleus [120,121]. Deterioration of the nuclear envelope determines the activation of cyclic GMP-AMP (cGAS) and the downstream secondary activation of stimulator of interferon genes (STING), a sensor of cytoplasmic dsDNA and a potent inducer of the senescence-associated secretory phenotype (SASP) [121], making YAP/TAZ an important inhibitor of senescence through these effects.

Inflammation, mechanical overload and senescent features interfere at the level of ADAMTS5, the major cartilage aggrecanase. RunX2, NFkB, YAP/TAZ all control the expression of ADAMTS5. Mechanical inputs act through the canonical, but also hippo-independent, YAP/TAZ signaling and co-operate with RunX2 to mediate hypertrophy [122]. Zhang et al. observed that YAP expression and localization of rat chondrocytes differed on fibronectin-coated polyacrilamide gels of different stiffness [123]. On stiff substrates, YAP had higher expression and nuclear localization, whereas on soft, more elastic substrates, YAP was less expressed and localized to the cytoplasm [124]. The authors also described that LATS1 expression of chondrocytes grown on hard substrates was extremely low, and LATS1 knockdown was associated with low expression of Sox-9, collagen type II and aggrecan. On soft substrates and *Lats1* knock-down, this shift in gene expression did not occur [124]. The stiffness of mineralized collagen-glycosaminoglycan matrices regulates osteoblastogenesis from mesenchymal stem cells, but this observation has not been translated to chondrocytes.

These experiments show that YAP/TAZ deficiency is associated with an overt osteoarthritic phenotype and demonstrate that YAP expression is coupled to mechanosensing and determines chondrocyte differentiation, partly in a Hippo pathway-dependent manner. CRISPR/Cas-9-mediated knockdown of YAP resulted in premature senescence of mesenchymal stem cells due to forkhead box D1 (FOXD1) inactivation, whereas YAP overexpression reversed these effects [125]. Targeting the YAP1/p21 axis has been proposed as a promising senolytic approach [123].

Hippo pathway players adapt to oxidative conditions Nucleus pulposus mesenchymal stem cells used for experimental treatment of disc degeneration showed increased proliferative capacity and maximal inhibition of Hippo signaling when treated with 75 μg H_2_O_2_. In addition, H_2_O_2_ pretreatment induced a strong repair capacity in nucleus pulposus mesenchymal stem cells, which developed an efficient antioxidant response as reflected by decreased Bromodomain-containing protein (Brd4) and Kelch-like ECH-associated protein 1 (Keap1), but increased leucine zipper Nrf2 expression [126]. An opposite effect on these proteins was observed when high concentrations of H_2_O_2_ (300 μM) were applied [126].

Under experimental conditions, oxidative stress induces the acetylation of MOB1-K11 at lysine 11 by CBP, thus counteracting its ubiquitination. MOB1-K11 acetylation is followed by phosphorylation and activation of LATS. This mechanism is protective against tumor growth, which is further supported by the observation that the acetylation-deficient mutant MOB1-K11R promotes tumor cell proliferation, migration and invasion [127]. Interestingly, mechanical stress has different effects on YAP/TAZ depending on its nature: turbulent vascular flow has a stimulatory effect, promoting nuclear translocation of the complex, endothelial proliferation and vascular inflammation, whereas laminar flow lacks these consequences [128]. If YAP/TAZ acts as a fine-tuned mechanosensor in chondrocytes, it may also mediate high-level adaptation to different levels of shear stress in chondrocytes.

Toxic substances such as fluoride exerted multiple effects on ATDC5 chondrocytes: increased oxidative stress, inhibited Hippo signaling and increased extracellular matrix degradation. A total of 20 mg/L sodium fluoride (NaF) increased malondialdehyde concentration, suppressed total antioxidant capacity, glutathione peroxidase and superoxide dismutase activity. Among the Hippo pathway players, higher levels of YAP-1 but lower levels of p-MST1/2, p-LATS1/2 and p-YAP1 were detected. In addition, COL2A1 and aggrecan expression were significantly lower and MMP-13 expression was higher compared to controls [129]. Some observations contradict these results: silencing of YAP by siRNA abolishes the IL-1β-induced catabolic enzyme signature [130].

The YAP/TAZ activation, main cellular effects and interferences with inflammation are synthesized in Figure 2.

The transcriptional regulator YAP/TAZ might control the progression of OA at crucial levels. It inhibits chondrocyte differentiation, maintains Sox-9 activity, COL2A1 and ACAN synthesis, and suppresses MMP-13. These results were confirmed in LATS1 and MST1/2 knock-out models with YAP overexpression. An important effect of YAP/TAZ is the protection of nuclear envelope integrity, the suppression of cGAS-STING danger signals and the resulting senescence-associated secretory pattern. YAP/TAZ is induced by high levels of oxidative stress and possibly by the variable shear stress of cartilage.

## 10. The Energy Sensor mTOR Protein Complex Regulates Chondrocyte Senescence

The mammalian target of rapamycin complex, composed of five subunits (mTORc, Raptor, MLST8, PRAC40, DEPTOR), is a complex energy and redox sensor [131] with an important role in monitoring protein translation and cellular processes dependent on protein synthesis. Pro- and anti-inflammatory cytokines, autophagy and the mTORc pathway interact in the regulation of cellular senescence. Bao J et al. used IL-18 to induce inflammatory conditions in rat chondrocytes and described the activation of the PI3K/Akt/mTOR signaling axis in parallel with the downregulation of COL2A1 and aggrecan transcription and translation. They also observed an increased apoptotic signature (Bax, Bcl2, Caspase3/9) and suppression of autophagy (Atg5 and 7, Beclin-1, LC3). These effects were reversed by specific inhibitors of PI3K, Akt and mTOR, in particular rapamycin [132].

Lu H. et al. treated primary wild-type mouse chondrocytes with tert-butyl hydroperoxide (TBHP) to induce oxidative stress and monitored the protective effects of fibroblast growth factor 21 (FGF21). TBHP induced catabolic effects on the target cells, decreased the synthesis of COL2A1 and aggrecan, but upregulated MMP-13 and ADAMTS-5; all these effects were reversed by administration of FGF21 [133]. In addition, FGF21 promoted autophagy flux and reduced the number of SA-β-gal-positive senescent chondrocytes, also suppressing p16INK4a and p21WAF1. These effects were mediated through the SIRT1-mTOR pathway, with FGF21 inducing SIRT1 but inhibiting phosphorylated mTOR; nicotinamide, a SIRT1 blocker, reversed the downstream protective effects of the cytokine [133].

Metformin, a classic drug used in the treatment of diabetes, was found to be protective in an OA model of medial meniscal destabilization. Metformin downregulated mTORc1 and p16INK4a in OA chondrocytes along with MMP-13 expression, and enhanced AMPK polarization in a dose-dependent manner [134].

Post-transcriptional regulation has also been documented: high circular RNA-7 (ciRS-7) and low micro-RNA-7 (miR-7) restore autophagy suppression in IL-1β-triggered chondrocytes via PI3K/Akt/mTOR signaling, thereby alleviating cartilage degradation. IL-1β and miR-7 mimics stimulate the intensity of IL-17A transcription and translation (a pro-inflammatory cytokine produced by Th17 cells and associated with arthritis), which is controlled and downregulated by ciRS-7 [135].

## 11. Regulation of Senescence at the Post-Transcriptional Level with miR and Other Small Non-Coding RNAs Modulates the Inflammatory Signature

MicroRNAs, as short non-coding oligo-ribonucleotides consisting of 20–24 subunits, are inhibitory sequences that couple to the 3′-noncoding sequences of specific target RNAs, modulating their stability and stopping their translation into proteins. Comparison of microRNA expression in osteoarthritic cartilage shows differential expression compared to healthy cartilage, but there is variability between groups, making interpretation difficult [136]. Asporin, a small leucine-rich proteoglycan with an essential role in senescence, enhanced osteoarthritis when overexpressed in mouse chondrocytes and its knockdown restored cellular homeostasis. miR-26b-5p, which is downregulated in OA, can suppress the effects of asporin by restoring TGF-β1-Smad2 signaling [137]. miR-33-5p is upregulated in senescent conditions and its specific inhibitor normalizes otherwise downregulated SIRT-6 expression. Cartilage degradation and chondrocyte senescence were induced when agomir-33-5p was applied intra-articularly in experimental animals, suggesting that miR-33-5p may be a therapeutic target [138]. Deletion of some microRNAs, such as miR-140, leads to cartilage degradation and susceptibility to post-traumatic OA [136]. miR-34a expression is increased in human OA; this molecule suppresses delta-like protein 1 (DLL1) and cell proliferation, leading to senescence and cell death [136].

Xiao F et al. found 40 microRNAs upregulated and 70 downregulated in senescent chondrocytes, of which inhibition of miR-132-5p prevented chondrocyte senescence [139]. Furthermore, 8 piRNAs were upregulated while 17 were downregulated, with a dominant species piR_025576 delaying chondrocyte senescence. There were also differences in the expression of 24 small nuclear RNAs (upregulated) and another 28 snoRNAs (downregulated). Metformin inhibited miR-34a, senescence-associated protein P16, IL-6 and MMP-13, whereas it increased SIRT-1, collagen type II and aggrecan expression in OA chondrocytes [140]. Mechanical overload reduced miR-325-3p and triggered the p21/p53 pathway, inducing chondrocyte senescence and lumbar facet joint degeneration [141].

Milk fat globule-epidermal growth factor (GF) factor 8 (MFG-E8), also known as lactadherin, reverses the progression of OA by preventing phosphorylation of the p65 subunit and thus inhibiting NF-kB. MFG-E8 strongly suppressed chondrocyte senescence, as evidenced by an increase in collagen type II protein and a significant decrease in P16 signaling and SA-β Gal and MMP-13 expression. MFG-E8 also downregulates the pro-inflammatory proteins IL-6 and TNFα but promotes an M2-type phenotype switch of macrophages. miR-99b-5p is an antagonist of these overt effects, and this microRNA is overtly upregulated in osteoarthritic cartilage [142].

miR-101 competes with the long non-coding RNA LINC00623 for the Harvey rat sarcoma viral oncogene homolog HRAS protein, a mediator of apoptosis and senescence (SA-β-Gal positivity) in IL-1β-treated chondrocytes [143]. miR-495 suppresses Akt and m-TOR phosphorylation in primary rat chondrocytes, human chondrocytes and SW1353 chondrosarcoma cells [144].

The key regulator of MMP-1 and MMP-13 synthesis and hallmark of senescence, p16INK4a, is downregulated by miR-24, as demonstrated by genome-wide miR analysis in IL-1β-treated primary and mesenchymal cell-differentiated chondrocytes [145].

## 12. Senolysis Has Beneficial Effects in Experimental Conditions

Chondrocyte SASPs may contribute to synovial inflammation by recruiting pro-inflammatory macrophages and T cells and downregulating the secretory capacity of fibroblasts. Osteocytes, subchondral bone marrow cells and adipocytes of the infrapatellar fat pad also show signs of senescence to some extent, and are further sources of a secreted inflammatory signature [146]. The selective elimination of senescent cells by immune response augmentation or senolytic drugs has become a potential therapeutic approach of great interest [92]. In p16-3MR transgenic mice, p16INK4A-positive and thymidine kinase-bearing senescent cells can be eliminated by the anti-herpesvirus drug ganciclovir [146]. In post-traumatic OA induced by ACLT surgery, the number of senescent chondrocytes and the expression of IL-1β, IL-6 and MMP-13 increase biphasically [92]. Senescent chondrocytes from total knee arthroplasty specimens could be eliminated by small molecules, and an increase in major collagen type II and aggrecan synthesis was observed [146]. For the effective treatment of OA, senolysis combined with general SASP inhibition in the joint could be a real breakthrough. Dendrobin, a bioactive alkaloid isolated from Dendrobium nobile, appears to be a promising candidate, as it inhibits extracellular matrix degradation by MMP-13, ADAMTS-5, NF-kB-mediated inflammation and mitochondrial ROS production, and enhances COL2A1 and ACAN gene expression [147]. An FDA-approved drug used to treat dyslipidemia also showed a dual senolytic and anti-inflammatory effect through PPARα signaling and increased autophagy in normal, ageing and osteoarthritic human chondrocytes [148]. Pterostilbene, an anti-inflammatory, antioxidant and anti-ageing compound, was shown to have potent anti-senescent activity, as its 5-week intraperitoneal administration improved Mankin and OARSI scores and inhibited the PI3K/Akt/NF-kB axis while decreasing the number of S-β-Gal-positive chondrocytes and the expression of p16, p21, IL-6, ADAMTS5 and MMP-13 [89]. Butorphanol, a sedative and analgesic agent with anti-inflammatory properties, had anti-senolytic activities in a HC-A cell culture model, decreasing TNF-α induced SA-β-Gal positivity and inhibiting IL-6 and IL-8 by decreasing NFkB and STAT3 phosphorylation [149]. Several miR species have been found to control key steps of senescent transformation. Zhu J. et al. performed inventive experiments to prepare an efficient miR-based therapeutic tool. They first found, via microRNA sequencing, that miR-29b-5p is decreased in primary osteoarthritic and OA mimetic, IL-1β-treated chondrocytes, and that this miR has protective effects in OA by silencing the ten-eleven translocation enzyme 1 (TET1) gene transcript, which is involved in OA progression [150]. miR29b-5p stimulates the synthesis of COL2A1, aggrecan and Sox-9 at both mRNA and protein levels, and inhibits a number of degradative enzymes (MMP-3, MMP-13, ADAMTS-4, and ADAMTS-5), p21 and p16lnk4a. A combined therapeutic tool based on cholesterol-modified 29b-5p agomir encapsulated in a hydrogel delivery system for sustained release and peptide repeats for stem cell recruitment proved to be very efficient when administered to rats 4 weeks after ACLT surgery [150,151].

## 13. Conclusions

Osteoarthritis shows various signs of low-grade, local, and in some aspects also systemic, inflammation, leading to prominent structural and functional changes in cartilage, synovial membrane and subchondral bone. There is a limited number of studies targeting or setting up models for very early OA; thus, it is difficult to establish a precise timeline of the initial trigger factors. In triggered cell culture, sera and synovial fluid of OA subjects, soluble mediators of inflammation, are clearly detectable, whereas the synovial membrane and articular chondrocytes show enhanced inflammatory signaling and high oxidative stress. The upregulation of NF-kB and a variety of pro-inflammatory cytokines is typically associated with catabolic processes and the overexpression of MMP-13, ADAMTS-4 and ADAMTS-5. The loss of proteoglycans, aggrecan and collagen type II content and the downregulation of Sox-9 are characteristic of matrix remodeling, while chondrocytes undergo hypertrophic and senescent transition. Hypertrophy, characterized by increased anaerobic glycolysis and the misregulation of key glycolitic enzymes (all functionally linked to the regulation of matrix homeostasis), also manifests in the overexpression of RunX2, metalloproteinases and several inflammatory mediators. Senescence is considered the end-stage hypertrophic differentiation stage, characterized by the increased expression of p16 and p21 and mitochondrial dysfunction, and has a strong inflammatory signature. The increased reactive oxygen species produced in the mitochondria of these transformed chondrocyte species may repress a recently discovered transcription factor complex, the Yes-associated protein, YAP, and its co-regulator, TAZ. These control chondrocyte differentiation, cartilage degradation and nuclear envelope integrity, thereby inhibiting senescence and its afferent inflammatory secretory pattern. Thus, YAP/TAZ may become an important pharmacological target and, together with senolytic agents, gain importance in future therapeutic strategies.

## Figures and Tables

**Figure 1 ijms-24-16468-f001:**
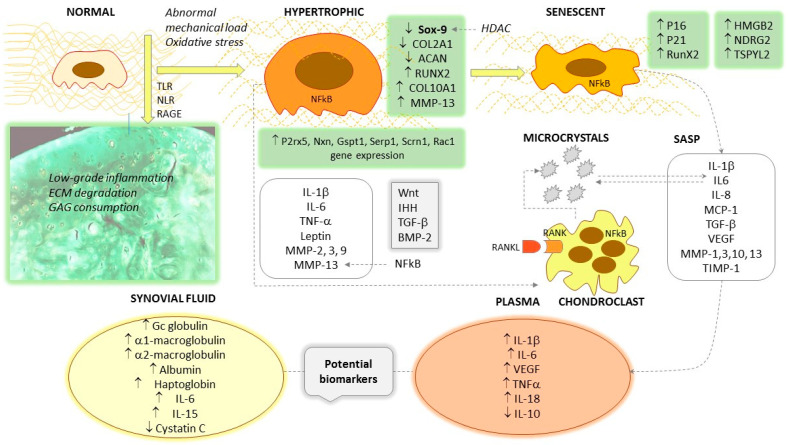
Hallmarks of the hypertrophic and senescent transformation of chondrocytes. ↑ symbolizes increased levels/stimulation; ↓ means decreased levels/inhibition.

**Figure 2 ijms-24-16468-f002:**
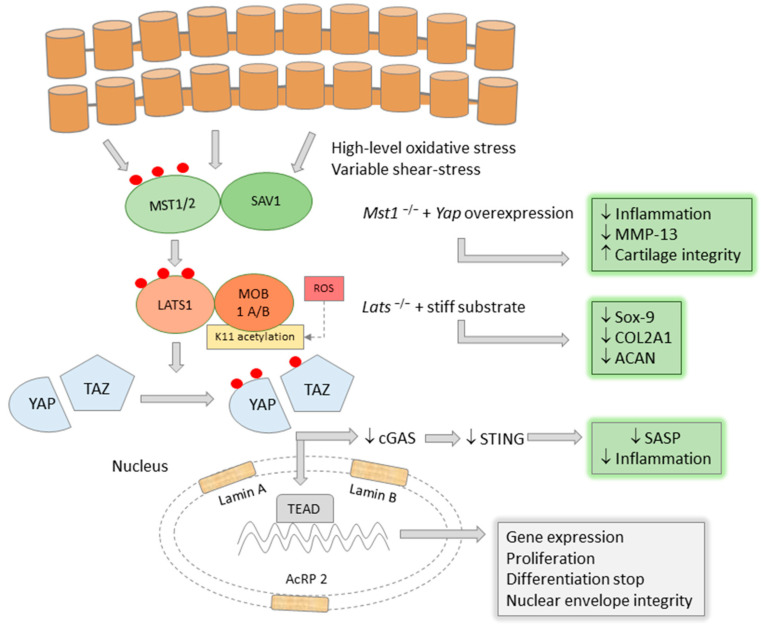
YAP/TAZ is a suppressor of inflammation and cartilage degradation.

**Table 1 ijms-24-16468-t001:** Circulating and histological markers of inflammation in human OA.

Reference	Study Goals	Change of Mediators	Other Results
Circulating mediators of inflammation			
Sohn D.H. et al. [8]	Proteomic analysis of serum/synovial fluid (SF)	Newly identified NF-kB-related proteins, cytokine receptors (IL-12R, IL-18R, IL-20R) and macrophage-derived inflammatory proteins in synovial fluid High levels of IL-6 in SF	Upregulation of histone deacetylase Upregulated plasma proteins, protease inhibitors, NF-kB subunits and regulators, cytokines (IL-6, MCP-1, VEGF), complement fragments in OA sera
Gobezie R. et al. [11]	Synovial protein analysis in OA patients and healthy	↑ Albumin, fibrinogen, α1-microglobulin/bikunin precursor, α2-macroglobulin, haptoglobin, complement C3 ↓ Cystatin C, aggrecan	18 Proteins differentially expressed between OA and healthy
Krenytska D. et al. [12]	Comparative analysis of plasma cytokines and growth factors in OA, OA + COVID-19 and controls	↑ IL-1β in OA ↓ TNF-α, NF-kB in OA ↓ VEGF, PDGF, FGF2 in OA No significant changes in IL-6 and HIF-1α	Maximum values of IL-1β, less pronounced decrease in TNF-α and NF-kB
Wang Z.W. et al. [14]	Serum protein analysis in OA patients	↑ IL-1β, IL-6, TNFα, VEGF in OA patients vs. controls	Increased expression of IL-1β, IL-6, TNFα, VEGF in synoviocytes
Stannus O. et al. [19]	Follow-up study of serum cytokines in an elderly cohort	Quartiles of IL-6 and TNF-α are associated with joint space narrowing	Baseline and changes of IL-6 predicts medial and lateral cartilage volume loss
Barker T. et al. [20]	Comparative study of serum IL-10 and TNF-α in different OA stages and controls	↓ IL-10 and IL-10/TNFα in subjects who underwent ACL or TJR surgery No significant changes in TNFα	IL-10 and IL-10/TNFα significantly differ between Kellgren–Lawrence scores 3 vs. 4 Low IL-10 suggests predisposition for development of severe knee OA
Panina S.B. et al. [21]	Comparative study of plasma and SF mediators in post-traumatic OA patients and controls	↑ plasma leptin, IL-1β and IL-6	Plasma leptin and SF IL-18 correlate with the Kellgren–Lawrence score in PTOA Significant correlation between plasma and SF leptin, IL-6 and IL-18
Waszczykowski M. et al. [22]	Comparative study of serum and SF cytokines in OA vs. healthy controls	↑ serum IL-6, IL-18 and IL-20 in OA vs. control group	IL-18 correlates with MMP-3 in OA patients ROC curve of IL-20 suggests diagnostic potential
Histological markers of inflammation			
Wang Z.W. et al. [14]	Synovial membrane immunohistochemistry analysis	↑ IL-1β, IL-6, TNF-α and VEGF synovial membrane expression in moderate/advanced OA compared to mild OA	↑ serum IL-1β, IL-6, TNFα, VEGF in OA patients vs. controls
Qu X.Q. et al. [25]	Comparative genetic and immunohistochemistry study of OA vs. normal cartilage specimens (knee arthroplasty and meniscus surgery)	↑ IL-6 and MMP-9 expression in OA specimens	↑ IL-6 and MMP-9 gene expression in OA cartilage
Warner S.C. et al. [27]	Immunohistochemistry and explant culture studies of OA patients	↑ IL-15Rα in OA samples	IL-15 treatment induction of MMP-1 and MMP-3
Scanzello C.R. et al. [28]	Synovial membrane immunohistochemistry in early vs. advanced OA patient cohort	↑ IL-15Rα in synovial membrane cells from patients with advanced OA	Increased IL-15 in SF, correlating with IL-6
Iannone F. et al. [29]	Comparative immunohistochemistry study of OA vs. healthy subjects	↑ IL-10 protein and mRNA expression in high-intensity cartilage lesions	No relationship between IL-10R expression and cartilage lesion degree
Vuolteenaho et al. [34]	OA patients cohort study of phenotype-explanted cartilage	↑ IL-6, IL-8, PGE_2_, Cox-2 expression in cartilage cultures obtained by joint replacement, induced by leptin/IL-1β	Selective inhibition of iNOS suppressed IL-6, IL-8, PGE_2_

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
