# Peer review of "Inflammatory and Metabolic Signaling Interfaces of the Hypertrophic and Senescent Chondrocyte Phenotypes Associated with Osteoarthritis"

_ijms, 2023, doi:10.3390/ijms242216468_

Round 1

Reviewer 1 Report

Comments and Suggestions for Authors

The “Inflammatory and metabolic signaling interfaces of the hyper-trophic and senescent chondrocyte phenotypes associated with osteoarthritis “ review is massive full of content ranging in many details on the Pathways and the aspects involved.

here below are some comments and considerations about missed nuances or imperfect wording:

-          lines 107-108: “Several factors and subunits of NF-kB….have been detected by proteomics…”, increased? Decreased? Altered ratio? Please specify.

-          Lines 154-155: “Serum leptin levels in obese OA patients are elevated compared to normal…” Please rewrite this sentence specifying what is meant for “normal”, not obese-not OA? Not obese-OA?
moreover, you could also add the concept that leptin induces the production of matrix metalloproteinases (MMPs), pro-inflammatory mediators and nitric oxide (NO) in chondrocytes.

-          Line 225: “Osteoarthritic cartilage shows aggregated islands of chondrocytes of variable shape” do you mean that the difference between superficial and lower is not flatter and rounder or globally is a messy and random shape?

-          Lines 371-375: when you are describing treatments with IL1b to induce inflammation, please also consider in vitro-indirect treatments as the use of conditioned medium (CM) of U937.

-          Line 392: Glucose-6-phosphate dehydrogenase (G6PDH), please recheck spaces and eventually hyphen.

-          Line 401: “IL-1b treatment and bone crystal microparticle…” please rephrase to be more specific, moreover consider introducing the indirect treatment (same comment of Lines 371-375).

-          Line 498: IL-1b-treated chondrocytes, please rephrase to be more specific,

-          Lines 401 and 499: bone crystal microparticle & Bone mineral particles, please add a sentence describing them.

-          Lines 700 to 731: is worth a mention also β-Caryophyllene treatments, there is a large literature on that comprehensive of in vivo and in vitro studies; moreover not only PPAR-alpha but also PPAR-gamma is involved “In Vitro Effects of Low Doses of β-Caryophyllene, Ascorbic Acid and D-Glucosamine on Human Chondrocyte Viability and Inflammation. Pharmaceuticals 2021”.

A recent study provided evidence that BCP significantly ameliorates the severity of arthritis in mice, reducing the expression of pro-inflammatory cytokines through a cross-talk between CB2 and PPAR-γ receptors “Picciolo, G.; Pallio, G.; Altavilla, D.; Vaccaro, M.; Oteri, G.; Irrera, N.; Squadrito, F. β-Caryophyllene Reduces the Inflammatory Phenotype of Periodontal Cells by Targeting CB2 Receptors. Biomedicines 2020, 8, 164.”

The conclusions are missing some considerations on the inflammation role, please consider adding a couple of sentences on the basis of previous comments.

Comments on the Quality of English Language

English loos nice

Author Response

Dear Reviewer,

Thank you for your careful revision and valuable comments.

The “Inflammatory and metabolic signaling interfaces of the hypertrophic and senescent chondrocyte phenotypes associated with osteoarthritis “ review is massive full of content ranging in many details on the Pathways and the aspects involved.
Here below are some comments and considerations about missed nuances or imperfect wording:

  1. Reviewer: lines 107-108: “Several factors and subunits of NF-kB….have been detected by proteomics…”, increased? Decreased? Altered ratio? Please specify.

Authors: Sohn et al. performed a protein mapping of 5 OA patients with Kellgren-Lawrence scores of 2 to 4. Synovial proteins were separated by PAGE and identified by liquid chromatography-tandem mass spectrometry. The data were not compared with other groups; however, a profile of 111 proteins was identified among which NF-kB subunits and NF-kB-related proteins, normally occurring in the intracellular space. We modified the manuscript as follows:

L110-113: “A polyacrilamide gel-liquid chomatography-tandem mass spectrometry proteomics approach identified the presence of several factors and subunits of NF-kB in synovial fluid, such as NF-kB-repressing factor, NF-kB inhibitor-like protein 1, NF-kB p100 subunit which are normally found only in the intracellular space.”

  1. Reviewer:Lines 154-155: “Serum leptin levels in obese OA patients are elevated compared to normal…” Please rewrite this sentence specifying what is meant for “normal”, not obese-not OA? Not obese-OA?
    Moreover, you could also add the concept that leptin induces the production of matrix metalloproteinases (MMPs), pro-inflammatory mediators and nitric oxide (NO) in chondrocytes.
    Authors: in the study of Lambova et al., serum leptin was significantly higher in OA patients, especially with high BMI, than in healthy controls, or those with OA and low BMI. We revised the manuscript as follows:

L149-L151: “Serum leptin levels are elevated in obese OA patients, especially those with a high body-mass index (BMI), compared to non-OA, non-obese, and osteoarthritic individuals with low BMI, and correlate with the radiographic stage of the disease.”

  1. Reviewer: Line 225: “Osteoarthritic cartilage shows aggregated islands of chondrocytes of variable shape” do you mean that the difference between superficial and lower is not flatter and rounder or globally is a messy and random shape?

Authors: L222-L223: “In contrast, osteoarthritic cartilage shows aggregated islands of chondrocytes of variable shape with the loss of characteristic organization and polarity”.

  1. Reviewer: Lines 371-375: when you are describing treatments with IL1b to induce inflammation, please also consider in vitro-indirect treatments as the use of conditioned medium (CM) of U937.
    Authors: we introduced the following text in the manuscript:

L380-L382: “Chondrocytes grown on U937 monocyte-derived macrophage-conditioned medium synthesize high-levels of IL-1b, IL-6, TNFa and the mitochondrial ROS. Hyaluronic acid, lactose-modified chitosan and their mixtures significantly reduced these effects.”

  1. Reviewer:Line 392: Glucose-6-phosphate dehydrogenase (G6PDH), please recheck spaces and eventually hyphen.

Authors: Thank you for this observation, we have corrected the denomination of the enzyme.

 L372: “…which is supported by the stimulation of glucose-6-phosphate dehydrogenase (G6PDH) activity in chondrocyte hypertrophy”.

  1. Reviewer: Line 401: “IL-1b treatment and bone crystal microparticle…” please rephrase to be more specific, moreover consider introducing the indirect treatment (same comment of Lines 371-375).

Authors: the effects of PMA and LPS conditioned U937 medium has been presented before.

  1. Reviewer: Line 498: IL-1b-treated chondrocytes, please rephrase to be more specific,

Authors: we rephrased this sentence:

“Primary mouse chondrocytes, cultured in DMEM, treated with 10 ng/mL recombinant IL-1b produce receptor activator of NF-KB (RANKL), which determines osteoclastogenesis in subchondral bone.”

  1. Reviewer: Lines 401 and 499: bone crystal microparticle & Bone mineral particles, please add a sentence describing them.

Authors: bone microparticles in this study have been obtained from long cortical porcine bones by mechanical shredding, sterilization and 70% alcohol wash.

L382-L385: “24 hours treatment with 10 ng/mL recombinant IL-1b and heterogeneic bone crystal microparticles (obtained from long cortical porcine bones by mechanical shredding, sterilization and 70% alcohol wash) also induced mitochondrial dysfunction, with increased ROS levels due to increased electron transport chain activity.”

  1. Reviewer: Lines 700 to 731: is worth a mention also β-Caryophyllene treatments, there is a large literature on that comprehensive of in vivo and in vitro studies; moreover not only PPAR-alpha but also PPAR-gamma is involved “In Vitro Effects of Low Doses of β-Caryophyllene, Ascorbic Acid and D-Glucosamine on Human Chondrocyte Viability and Inflammation. Pharmaceuticals 2021”.

Authors: we updated the manuscript as follows:

 L185-L189: “The overt linkage between inflammation and degradative pathways has also experimental evidence, lipopolysaccharide conditioning of chondrocyte culture media induced intensive NF-kB, IL-1b, and MMP-13 synthesis, associated with reduced viability. All of these expressions can be suppressed by antioxidant mixtures, such as ascorbic acid, b-carophyllene and D-glucosamine.”

  1. Reviewer: A recent study provided evidence that BCP significantly ameliorates the severity of arthritis in mice, reducing the expression of pro-inflammatory cytokines through a cross-talk between CB2 and PPAR-γ receptors “Picciolo, G.; Pallio, G.; Altavilla, D.; Vaccaro, M.; Oteri, G.; Irrera, N.; Squadrito, F. β-Caryophyllene Reduces the Inflammatory Phenotype of Periodontal Cells by Targeting CB2 Receptors. Biomedicines 2020, 8, 164.”

Authors: we have completed the manuscript with the following text:

b-carophyllene couples with the cannabinoid receptor CB2, and enhances the expressions of PPAR-g and PGC-1a, which may mediate its flagrant anti-inflammatory effects.”

11. Reviewer: The conclusions are missing some considerations on the inflammation role, please consider adding a couple of sentences on the basis of previous comments.

Authors: we completed the conclusions as follows:

L647-L649: “In triggered cell culture, sera and synovial fluid of OA subjects, soluble mediators of inflammation are clearly detectable, whereas the synovial membrane and articular chondrocytes show enhanced inflammatory signaling and high oxidative stress.

Reviewer 2 Report

Comments and Suggestions for Authors

The authors of this review have provided comprehensive understanding of how hypertophic and senescent chondrocytes and its pathways influence the process of osteoarthritis. This is a well-written review article and provides a good perspective on how hypertrophy and senescence results in osteoarthritis.

The authors of the article should consider the following points,

1.       In the introduction, there is no acknowledgement of recent evidence of cartilage progenitors cells with the cartilage tissue. This should be stated in the introductory paragraph and how are these cells not activated during the process of OA to prevent further cartilage damage ?

2.       The authors state the cytokine changes in OA serum and synovial fluid studies that show the increase in various inflammatory cytokines. Can the authors explicitly state the clinical grade (e.g. Kellgren-Lawrence, ICRS) score relative to the cytokine release in each of these studies within the text. Furthermore, a table with a summary of some of these studies would be helpful to the reader.

3.       How are current animal OA models useful in replicating human OA and is there a consensus on which model accurately represents the disease ? An opinion from the authors would be beneficial from the reader and/or table summarising the different models with their advantages and disadvantages should be provided.

4.       Based on the previous studies, what is the tissue that drives OA within the joint, is this the synovium, articular cartilage, subchondral bone or combination of all the tissues ? An opinion on this point, needs to be stated in the review.

5.       The receptor channel, TRPV4 has been implicated in the OA process from previous studies. How is or other cell surface or membrance channels implicated in the process of hypertrophy and senescence ?

6.       The focus of the metabolism section is on glucose pathways, how are fatty acids/lipids and amino acids also involved in the changes found in osteoarthritic chondrocytes and the resulting hypertrophic and senescence processes ? Recent studies have implicated leptin in the OA pathophysiology and this along with the stated metabolic molecules needs to be included.

7.       How are other cell types (i.e. immune cells and macrophages) involved in the senescence process during OA ? Description of other cell types in the etiology of OA would show the reader that this is a highly complex process and not solely involving chondrocytes.

Author Response

Thank you for your careful revision and valuable comments.

The authors of this review have provided comprehensive understanding of how hypertophic and senescent chondrocytes and its pathways influence the process of osteoarthritis. This is a well-written review article and provides a good perspective on how hypertrophy and senescence results in osteoarthritis.

The authors of the article should consider the following points,

  1. Reviewer: In the introduction, there is no acknowledgement of recent evidence of cartilage progenitors cells with the cartilage tissue. This should be stated in the introductory paragraph and how are these cells not activated during the process of OA to prevent further cartilage damage?

Authors: thank you for this interesting observation. We completed the manuscript as follows:

L59-L65: “Inefficient repair contributes to the progression of cartilage deterioration. Chondrogenic progenitor cells are overrepresented in osteoarthritic cartilage compared to the normal joint. In early OA, mobile cell clusters form near the erosions and fissures, showing both anabolic and catabolic phenotypes (Notch-1, STRO-1, VCAM-1). In advanced disease, migrating islands are distributed throughout the cartilage and can even pass through the fractured tidemark as shuttles between cartilage and subchondral bone. Their putative reparative role during disease progression is not fully understood and functional characterization of their subtypes is required to exploit their therapeutic potential.”

  1. Reviewer:The authors state the cytokine changes in OA serum and synovial fluid studies that show the increase in various inflammatory cytokines. Can the authors explicitly state the clinical grade (e.g. Kellgren-Lawrence, ICRS) score relative to the cytokine release in each of these studies within the text. Furthermore, a table with a summary of some of these studies would be helpful to the reader.

Authors: we introduced Table 1 with relevant human studies showing significantly changed soluble and histological marrkers of inflammation. Where it is the case, the results show also the Kellgren-Lawrence scores. Please, see Table 1, L157.

  1. Reviewer:How are current animal OA models useful in replicating human OA and is there a consensus on which model accurately represents the disease? An opinion from the authors would be beneficial from the reader and/or table summarising the different models with their advantages and disadvantages should be provided.

Authors: thank you for this observation. We introduced the following text into the manuscript:

L99-L102: “In general, animal models support the presence of pro-inflammatory factors in OA, but in some cases there is still some controversy. Experimental OA reflects both primary and secondary, post-traumatic human disease, and includes spontaneous (naturally occuring and on genetic background), induced (provoked surgically or chemically) and non-invasive murine, canine and lapine models applying a transarticular impact.

  1. Reviewer:Based on the previous studies, what is the tissue that drives OA within the joint, is this the synovium, articular cartilage, subchondral bone or combination of all the tissues? An opinion on this point, needs to be stated in the review.

Authors: the specific etiological factors in OA are multiple and the sequence of changes is an important basic question, yet remaining to be solved. Perhaps the five different clinical forms have different timelines and sequences. However, the discussion of this issue is beyond the scope of our review. We focused on changes related to the hypertrophic and senescent chondrocyte phenotype, and their relationships with the inflammatory pathway. We introduced the following sentences in the Introduction section:

L53-L56: “The early events in osteoarthritis are likely to involve the synovial membrane, which is well-vascularized, has mechanoreceptors and is therefore sensitive to systemic effects. In parallel or in rapid succession, abnormal mechanical loading and other factors may also result in chondrocyte phenotype switching and matrix remodeling”

  1. Reviewer:The receptor channel, TRPV4 has been implicated in the OA process from previous studies. How is or other cell surface or membrane channels implicated in the process of hypertrophy and senescence?

Authors: TRPV4 has been identified by functional gene screening as regulator of chondrogenic differentiation. The Transient Receptor Potential Vanilloid 4 is an ion channel taking a pivotal role in cartilage mechanotransduction and regulation of inflammatory pathways. In the absence of direct evidence, we suppose that TRPV4 may interfere with the hypetrophic or senescent transition at least through 4 mechanisms: a.) regulating the infiltrating macrophage phenotypes, b.) regulating the expression of IL-1R and the subsequent IL-1 signaling, c.) suppressing NF-kB, d.) regulating Sox-9. Taking these into consideration, we introduced the following sentences into the manuscript:

L170-L172: “The Transient Receptor Potential Vanilloid 4 (TRPV4), an ion channel identified by functional gene screening to play a pivotal role in chondrocyte mechanotransduction, stimulates Ca2+ influx, Sox-9, and matrix synthesis. TRPV4 is a regulatory checkpoint that also switches off also the IL-1 signaling through down—regulation of IL-1R].

  1. Reviewer:  The focus of the metabolism section is on glucose pathways, how are fatty acids/lipids and amino acids also involved in the changes found in osteoarthritic chondrocytes and the resulting hypertrophic and senescence processes? Recent studies have implicated leptin in the OA pathophysiology and this along with the stated metabolic molecules needs to be included.

Authors: the main chemical fuel of chondrocytes is glucose. Although there is little evidence for the direct implication of lipids and amino acids in chondrocyte hypertrophy and/or senescence, in some aspects these metabolically interfere with inflammation, cartilage degeneration or regeneration. We introduce several sentences in the manuscript, as follows:

“If chondrocytes derive their energy mainly from glucose, with regard to alternative nutrients, little is known about the direct effects of lipid and amino acid metabolism on hypertrophy and/or senescence. However, these substances interfere with inflammation, cartilage degeneration and remodelling at several checkpoints. First, there is impaired  oxidation in the mitochondria and a reduced carnitine shuttle, which is essential for the transfer of various activated acyl groups. Saturated fatty acids such as stearate and palmitate are pro-inflammatory, the former increasing cytokine production via HIF-1a and the latter inducing Cox-2 and IL-6 [73]. N-6 polyunsaturated fatty acids increase ROS production by stimulating NADPH+H+ oxidase 4, promote apoptosis, and arachidonic acid contributes to the generation of pro-inflammatory prostaglandins and leukotrienes [73]. N-3 polyunsaturated fatty acids are strongly anti-inflammatory, as they form pro-resolving mediators such as resolvins, protectins and maresins, and suppress ADAMTS-4, ADAMTS-5, MMP-3 and MMP-13 [83]. In addition, osteoarthritic chondrocytes increase their cholesterol uptake and produce increased amounts of oxysterols [84]. Phospholipid degradation products, such as lisophospholipids produced by phospholipase A2 (PLA2), recruit phagocytes into the joint. Many quantitative changes in amino acids have been proposed as biomarkers, such as an increase in arginine/asymmetric dimethylarginine or a decrease in the ratio of branched-chain amino acids/histidine. In synovial fluid, glutamine, threonine and nitrotyrosine reach high concentrations; increased proteolysis and nitrosative stress may contribute to these metabolic changes [73]. Arginase 2, an important enzyme of infiltrating macrophages, upregulates MMP-3 and MMP-13 via NF-kB [85].”

Further, we mentioned leptins’ role in the below section:

“One possible trigger of the cyclooxygenase pathway is leptin: a concentration of 10μg/mL increased the expression of Cox-2 and IL-6 as well as MMP-1, -3, -13, whereas suppressor of cytokine signalling-3 (SOCS-3) was able to stop these effects [32]. Serum leptin levels are elevated in obese OA patients, especially those with a high body-mass index (BMI) ,compared to non-OA, non-obese, and osteoarthritic individuals with low BMI, and correlate with the radiographic stage of the disease [33]. Moreover, 10 μg/mL leptin triggering alone, or with simultaneosly administered IL-1b (10pg/mL) stimulated the expression of inducible nitric oxide synthase, PGE2, IL-6 and IL-8 in femoral condyli and tibial plateau cartilage specimens obtained on the occasion of joint replacement [34]”

  1. Reviewer:   How are other cell types (i.e. immune cells and macrophages) involved in the senescence process during OA? Description of other cell types in the etiology of OA would show the reader that this is a highly complex process and not solely involving chondrocytes.

Authors: In many phrases and sections, we mentione the involvement of macrophages, synovial fibroblasts and subchondral cells (osteoblasts, osteocytes and osteoclasts). Chondrocytes are relatively isolated in the cartilage, and are exposed primarily to soluble mediators, microenvironment factors, mechanical and oxidative stress. Treating the full complexity of cellular interactions in osteoarthritis, and their possible effect on chondrocyte senescence is beyond the scope of this review.

Round 2

Reviewer 2 Report

Comments and Suggestions for Authors

The authors have addressed my concerns appropriately.